# Research on the Pavement Performance of Slag/Fly Ash-Based Geopolymer-Stabilized Macadam

**Jinchao Yue [1], Xiaofan Nie [1], Ziran Wang [1], Junlei Liu [2] and Yanchun Huang [3,\*]**

1 Yellow River Laboratory, Zhengzhou University, Zhengzhou 450001, China
2 Zhengzhou Municipal Engineering Survey Design & Research Institute, Zhengzhou 450003, China
3 School of Physical Education, Zhengzhou University, Zhengzhou 450001, China
* Correspondence: ych999@zzu.edu.cn; Tel.: +86-135-2340-1340

**Abstract:** The substitution of slag-based geopolymer materials for cement-stabilized macadam in road bases is in line with the demand for green and sustainable development in the transportation industry. Thus, slag/fly-ash-based geopolymer materials were prepared to stabilize macadam of road bases in this study. The mechanical properties, freeze-thaw resistance, and dry shrinkage durability of slag/fly-ash-based stabilized macadam materials were studied to analyze the influence of geopolymer dosage on these properties of stabilized macadam. Microscopic tests such as XRD, FITR, and SEM were carried out to explore the formation mechanism of strength and the characteristic of interface transition zone (ITZ). Results show that the 28 d compressive strength, compressive elastic modulus, and tensile strength of slag/fly-ash-based stabilized macadam increase linearly with the increase of geopolymer content. When the dosage of slag-based geopolymer is 4%, the 7 d unconfined compressive strength of slag/fly-ash-based stabilized macadam reaches 8.76 MPa, and the strength still reaches 14.84 MPa after five freeze-thaw cycles (28 d), which satisfy the application requirements of expressway and first-class highway base (JTG/T F20-2015). The dry shrinkage property of slag/fly-ash-based stabilized macadam is better than that of cement-fly-ash-stabilized macadam. When the amount of geopolymer is 3%, the dry shrinkage strain of slag/fly-ash-based stabilized macadam is 231.2 με, which is smaller than that of cement-fly-ash-stabilized macadam (261.3 με). No obvious porosity around the ITZ was detected, indicating good binding between the geopolymer and the aggregate.

**Keywords:** slag; fly ash; geopolymer; semi-rigid road base; pavement performance

## 1. Introduction

Cement and lime have been widely used in infrastructure as conventional binding materials for earth and rocks. However, their production consumes a lot of mineral resources, resulting in environmental pollution and excessive emissions of $CO_2$. At the same time, the secondary development and reuse of large quantities of industrial waste can provide a lot of raw materials for infrastructure construction. Geopolymer is an important type of nonmetallic cementitious material prepared by natural Si-Al materials or industrial wastes such as slag, fly ash, and/or steel slag [1–3]. Compared with traditional Portland cement, geopolymer exhibit excellent characteristics, including early strength, high strength, low $CO_2$ emissions, excellent corrosion resistance, and durability [3–5]. Therefore, many researchers use geopolymers to prepare mortar and concrete and strive to improve their properties.

Temuujin et al. [6] used low-calcium fly ash as raw material to prepare geomeric, and by adding CaO and $Ca(OH)_2$ to the raw material system, the 28 d mechanical strength of the product increased from 11.8 MPa to 29.2 MPa. Kallempudi et al. [7] carried out indoor experiments and found that with increased activator concentration and maintenance temperature, the compressive strength of low-calcium fly-ash-based geopolymer mortar was increased. Singh et al. [8] studied the influence of activator concentration on the characteristics of fly-ash/slag-based geopolymer concrete. It was found that the development

of compressive strength was earlier in geopolymer concrete than in ordinary Portland cement, and that it formed satisfactory bond with aggregates. Furthermore, with increased curing time, zeolite phase products played the leading role, resulting in minor drying shrinkage. Lahoti et al. [9] studied the main factors that influenced the compressive strength of metakaolin-based geopolymer, which supports that the Si/Al (mole ratio) is the most important parameter, followed by Al/Na (mole ratio). The water cement ratio has less influence in the compressive strength of metakaolin-based geopolymer, differing from ordinary Portland cement. Wang et al. [10] studied the change of fluidity and mechanical strength of geopolymers prepared with various water cement ratios. Results showed that the fluidity of geopolymers improved, but the mechanical strength decreased as the water cement ratio increased.

Geopolymers are expected to be suitable for serving as a potential cementitious material for stable aggregate base of pavements. According to the Detailed Rules for Highway Base Course Construction Techniques (JTG/T F20-2015), it is required that the stabilized macadam base course shall meet the requirements such as adequate strength, water stability, small shrinkage and deformation and strong resistance to scouring. Therefore, in addition to mechanical properties of geopolymers, many scholars also studied the aggregate adhesion and volume shrinkage of geopolymers. In geopolymer concrete, there is a transition area between slurry and aggregates [11–14], which controls the strength, resistance to chemical corrosion, permeability and other macro performance of geopolymer concrete [15,16]. Demie et al. [14] found that both wet curing and early-stage temperature rise curing can improve the transition area to obtain stronger adhesion. Khan et al. [15] found that the increased NaOH concentration in alkali-activator brough high solubility of aluminosilicate raw material, and that as a result, the micro structure of the transition area of interface was improved with reduced thickness. Lee and Van Deventer [16] found that when steel slag coarse aggregates were used, the Ca and Mg in aggregates participated in chemical reaction of geopolymers, which brought denser transition areas of interface and stronger adhesion. Lloyd et al. [17] found that the use of soluble silicate in alkali-activator could effectively improve the adhesion of interface. In addition, there are obvious shrinkage cracks of geopolymers during curing [18,19]. Excessive shrinkage will result in structural cracks, which can reduce the strength, rigidity, and service life of the structure [20–22]. Research shows that due to the filling of micro aggregates in fly ash, the increased dosage of fly ash can significantly reduce the dry shrinkage of geopolymers [23,24]. Duan et al. [25] believed that the addition of $TiO_2$ nanoparticles could improve the carbonization resistance and reduce dry shrinkage of geopolymers.

At present, there are few reports on the application of slag/fly-ash-based geopolymers in macadam base, and thereby further research is required. In this paper, slag/fly-ash-based materials are used to substitute the cement binder in the stabilized macadam of road base. The mechanical property, frost-resistance, and dry shrinkage of this geopolymer material are studied to explore the feasibility of the application in semi-rigid base course, expecting to offer a certain technical guidance for the popularization and application of slag/fly-ash-based geopolymer in road engineering.

## 2. Experimental

### 2.1. Materials

The slag-based geopolymer material (Xi'an Changda Road Maintenance Company, Xi'an, China) used in the test is composed of slag and alkaline activator, among others. The specific surface area of the slag is 458.3 m$^2$/kg. The main chemical components of the slag are shown in Table 1. Alkaline activator is crystalline powder with other additives. The main active components include $SiO_2$, $Al_2O_3$, and $CaO$, and the specific surface area is 444.8 m$^2$/kg.

42.5 grade ordinary Portland cement (Pingdingshan Tianrui Cement Co., Ltd., Pingdingshan, China) and first-grade fly ash (Henan Borun Foundry Materials Co., Ltd., Xinyang, China) were used in the test. The main chemical components are shown in Table 1.

The crushing value of the macadam was 20%. The skeleton-density grading (JPA) is adopted for each test piece (JTG D50-2017), as shown in Table 2. In order to ensure the consistency of grading of each test piece, the stones were screened using a square-hole sieve according to the grading requirements prior to weighed and mixed.

**Table 1.** Chemical composition of raw materials (wt.%).

| Material | CaO | $SiO_2$ | $Al_2O_3$ | $Fe_2O_3$ | MgO | $SO_3$ | $K_2O$ | $P_2O_5$ | $TiO_2$ | $Na_2O$ |
|---|---|---|---|---|---|---|---|---|---|---|
| Slag | 64.61 | 17.57 | 5.01 | 4.15 | 3.04 | 2.70 | 1.31 | 0.42 | 0.35 | 0 |
| Fly ash | 4.02 | 53.90 | 31.15 | 4.16 | 1.01 | 0.73 | 2.05 | 0.67 | 1.13 | 0.89 |
| Cement | 64.44 | 21.60 | 4.13 | 4.57 | 1.06 | 0.13 | 0.65 | 2.01 | 0.67 | 0.11 |

**Table 2.** Grading result of macadam.

| Aperture Size/mm | 26.5 | 19 | 9.5 | 4.75 | 2.36 | 0.6 | 0.075 |
|---|---|---|---|---|---|---|---|
| Pass rate/% | 100 | 79 | 49 | 30 | 19 | 9 | 2 |

*2.2. Mix Proportions of Geopolymer-Stabilized Macadam*

The proportions are shown in Table 3. Briefly, 2% slag-based geopolymer (GFA-2), 3% (GFA-3), 4% (GFA-4), and 5% (GFA-5) specimens were selected for comparison with 3% cement (CFA-3) and 4% (CFA-4) specimens in the control group, and all proportions were expressed by mass ratio (%).

**Table 3.** The proportions of geopolymer-stabilized macadam.

| Samples | Stabilizer | Dosage of Geopolymer (%) | Dosage of Cement (%) | Dosage of Fly Ash (%) | Ratio of Binder to Stabilized Material (%) | Gradation Type |
|---|---|---|---|---|---|---|
| CFA-3 | Cement/fly ash mixture | - | 3 | 12 | 3:12:85 | JPA |
| CFA-4 | Cement/fly ash mixture | - | 4 | 16 | 4:16:80 | JPA |
| GFA-2 | Geopolymer/fly ash mixture | 2 | - | 8 | 2:8:90 | JPA |
| GFA-3 | Geopolymer/fly ash mixture | 3 | - | 12 | 3:12:85 | JPA |
| GFA-4 | Geopolymer/fly ash mixture | 4 | - | 16 | 4:16:80 | JPA |
| GFA-5 | Geopolymer/fly ash mixture | 5 | - | 20 | 5:20:75 | JPA |

*2.3. Preparation of Geopolymer-Stabilized Macadam*

Put the weighed fly ash and crushed stone aggregates into the mixer according to the mixing ratio and stir for 180 s to fully mix the materials; dissolve the slag-based geopolymer powder in water, pour it into the mixer, and stir for 180 s; pour the mixture into the 150 mm × φ150 mm, 100 × 100 × 400 mm mold. After initial setting, the concrete test block was removed from the mold and placed in a standard curing box with temperature of (20 ± 3) °C and relative humidity >90% for curing. After 7 days, it is taken out for subsequent testing.

*2.4. Testing Methods*

2.4.1. Compaction Test

In accordance with regulations for tests of stabilization materials of organic binders in Highway Engineering (JTG E51-2009), heavy compaction test was carried out on 6 groups

of stabilized pellets with different match ratio (CFA-3, CFA-4, GFA-2, GFA-3, GFA-4, GFA-5) and five samples with different water content (5%, 5.5%, 6%, 6.5%, 7%) were prepared for each group. The corresponding dry density with different water content was determined by weighing. The optimal water content ($\omega_0$) and corresponding maximum dry density ($\rho_d$) were determined by fitting prior to preparing test pieces.

### 2.4.2. Mechanical Properties Test

Test pieces with a size of 150 mm $\times$ $\varphi$150 mm are prepared for compressive strength and indirect tensile strength tests after 7 d, 28 d, and 60 d of curing, conducted on a system of microcomputer-controlled press machine. The compressive strength was determined by Equations (1) and (2), and the indirect tensile strength was determined by Equation (3), where $R_c$ is the unconfined compressive strength; $R_i$ is the indirect tensile strength; $P$ is the maximum destructive pressure of samples; $A$ is the cross-sectional area of samples; $D$ is the diameter of the test piece; and $h$ is the height of the test piece after immersion in water.

$$R_c = \frac{P}{A} \tag{1}$$

$$A = 0.25\pi D^2 \tag{2}$$

$$R_i = 0.004178\frac{P}{h} \tag{3}$$

The compressive elastic modulus test was carried out on samples with a size of 150 mm $\times$ $\varphi$150 mm using the top surface method. The predetermined unit pressure of 1.5 MPa was divided into five increments as the pressure value applied each time. According to the rule where loading pressure is increased gradually by one increment at each level, the process is repeated one level after another until the last group. The compressive elastic modulus under the load of each level was determined by Formula (4), where $E_c$ is the compressive elastic modulus; $P$ is the unit pressure; $h$ is the height of the test piece; and $l$ is the elastic deformation of the test piece.

$$E_c = \frac{Ph}{l} \tag{4}$$

### 2.4.3. Freeze-Thaw Test

Freeze-thaw test was carried out for slag/fly-ash-based geopolymer-stabilized macadam with four different dosages (GFA-2, GFA-3, GFA-4, GFA-5) to compare with cement-fly-ash stabilized macadam (CFA-4). Freeze-thaw tests were set to five cycles for samples after 28 d of curing. Test results are the average of three test pieces (150 mm $\times$ $\varphi$150 mm) were measured before and after free-thaw test. The loss of compressive strength (BDR) is determined by Formula (5), where $BDR$ is the loss of compressive strength after $n$ cycles of freeze-thaw; $R_{DC}$ is the compressive strength of the test piece after $n$ cycles of freeze-thaw; $R_C$ is the compressive strength of the control test piece.

$$BDR = \frac{R_{DC}}{R_C} \tag{5}$$

### 2.4.4. Dry Shrinkage Test

The dry shrinkage test was carried out for beam specimens with a size of $100 \times 100 \times 400$ mm. The length and initial mass ($m_0$) of specimens were measured after 7 d of curing. After that, the specimen was moved into a drying chamber, where the temperature was controlled at $20 \pm 2$ °C and the relative humidity was $50 \pm 5\%$. After 30 d of observation and recording, the specimen was placed into an oven for drying until a constant mass ($m_p$) was reached. The rate of water loss is determined by Formula (6); the drying shrinkage strain is determined by Formula (7); the drying shrinkage coefficient is determined by Formula (8); and the average drying shrinkage coefficient is determined by Formula (9), where $W_i$ is

the *i*-th water loss rate; $m_0$ is the initial mass of the test piece; $m_i$ is the *i*-th weighed mass of the test piece; $m_p$ is the constant mass of the test piece after drying; $\delta_i$ is the i-th drying shrinkage; $\varepsilon_i$ is the *i*-th drying shrinkage strain; and *l* is the length of the test piece.

$$\omega_i = \frac{m_o - m_i}{m_p} \tag{6}$$

$$\varepsilon_i = \frac{\delta_i}{l} \tag{7}$$

$$\alpha_{di} = \frac{\varepsilon_i}{\omega_i} \tag{8}$$

$$\alpha_d = \frac{\sum \varepsilon_i}{\sum \omega_i} \tag{9}$$

2.4.5. Microscopic Test

KYKY-EM6200 scanning electron microscope (SEM) is used to analyze the microscopic appearance of the bond between geopolymer and aggregate, Fourier transform infrared spectroscopy (FTIR, JASCO FT/IR-6100, FRITSCH GmbH, Idar-Oberstein, Germany) is used to analyze the molecular structure of geopolymer and chemical bond changes, X-ray diffraction (XRD, Panalytical X'pert Pro, Malvern Panalytical Ltd., Malvern, UK) is used to analyze the changes in the crystal structure of the geopolymer and the principle of strength generation.

## 3. Results and Discussion

### 3.1. Compaction Test

The optimal water content and maximum dry density under different geopolymer content are shown in Table 4. The maximum dry density of geopolymer decreases, while the optimal water content increases as the cementitious material increased. This is because voids of stones are completely filled up by the increased cementitious material, which reduces the demand of stones relatively. More water is needed for uniform mixing and compaction as the content of cementitious material increased because cementitious materials have a much higher absorption capacity than that of stones. The dry density of stabilized macadam decreased when the content of cementitious materials is higher than 15% due to the relatively low density of the cementitious material, and when there is less cementitious material, it cannot completely fill up the voids of the stones, resulting in relatively smaller dry density.

**Table 4.** Results of heavy compaction test of geopolymer-stabilized macadam.

| Code of Mix Proportion | Dosage of Geopolymer (%) | Dosage of Fly Ash (%) | Optimal Water Content (%) | Maximum Dry Density (g/cm³) |
|---|---|---|---|---|
| CFA-3 | 3 | 12 | 5.76 | 2.36 |
| CFA-4 | 4 | 16 | 6.17 | 2.32 |
| GFA-2 | 2 | 8 | 5.31 | 2.30 |
| GFA-3 | 3 | 12 | 5.82 | 2.41 |
| GFA-4 | 4 | 16 | 6.14 | 2.29 |
| GFA-5 | 5 | 20 | 6.49 | 2.23 |

### 3.2. Mechanical Tests

3.2.1. Influence of Curing Age on Compressive Strength and Elastic Modulus

The effect of curing age on compressive strength and elastic modulus of slag/fly-ash-based geopolymer-stabilized macadam (GFA-4) and cement-fly-ash-stabilized macadam (CFA-4) is shown in Figure 1a,b. It can be seen that the strength and elastic modulus of cement and geopolymer-stabilized macadam increased with curing ages. The compressive

strength and elastic modulus of geopolymer-stabilized macadam showed a sharply increase in the first 14 d of curing, and then increased gently. The compressive strength increases from 8.76 MPa to 13.2 MPa, an increase by 50.7% in the age range from 7 d to 14 d, which is accompanied by an increase of the elastic modulus from 2220 MPa to 2912 MPa (31.1%). Furthermore, the strength and elastic modulus of geopolymer stabilized macadam are higher than those of cement-stabilized macadam.

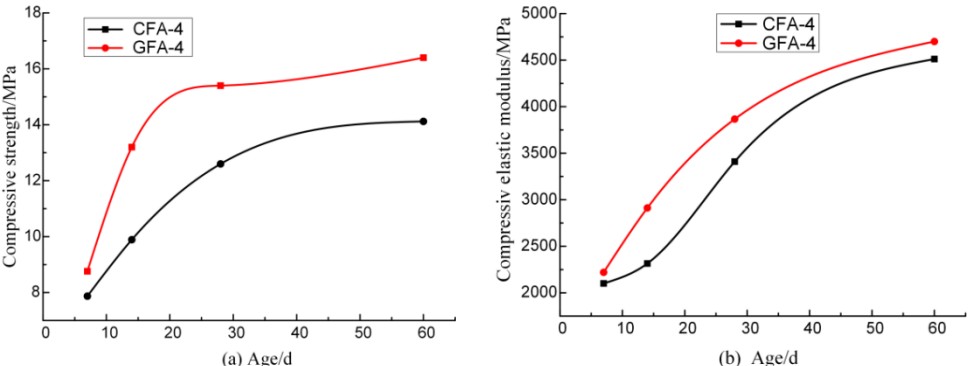

**Figure 1.** Mechanical properties change with age. (**a**) Compressive strength. (**b**) Compressive elastic modulus.

### 3.2.2. Influence of Geopolymer Dosage to Compressive Strength, Elastic Modulus, and Tensile Strength

Effects of cementitious material dosage on compressive strength, elastic modulus, and tensile strength of slag/fly-ash-based geopolymer stabilized macadam (28 d) is shown in Figure 2a–c, respectively. It can be seen that the dosage of cementitious material has a serious influence on the mechanical properties of the test piece. The compressive strength, elastic modulus, and tensile strength of slag/fly-ash-based geopolymer-stabilized macadam increase linearly with the dosage of cementitious material, which is consistent with the change of the compressive strength of polypropylene fiber-cement-stabilized macadam with cement dosage [26]. For example, when the dosage of cementitious material increases from 2% to 5%, the compressive strength increases by 69.5% (from 9.85 MPa to 16.70 MPa), with an increase by 49.4% for elastic modulus (from 2845 MPa to 4250 MPa), and the tensile strength increased by 89.2% (from 0.93 MPa to 1.76 MPa).

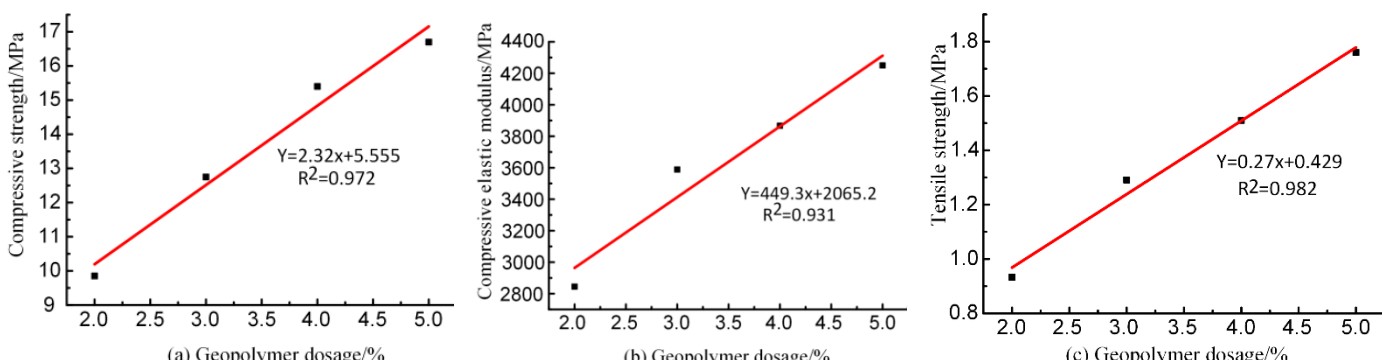

**Figure 2.** Effect of different geopolymer dosage on mechanical properties of specimens. (**a**) Compressive strength. (**b**) Compressive elastic modulus. (**c**) Tensile strength.

### 3.3. Freeze-Thaw Cyclic Test

The 28 d (five cycle) compressive strength loss (BDR) of slag/fly-ash-based geopolymer-stabilized macadam are shown in Table 5. It can be seen that the BDR value of geopolymer-stabilized macadam is greater than that of the cement-fly-ash-stabilized macadam under the same mix proportion after freeze-thaw cyclic tests. This suggests that the geopolymer

cementitious material have a higher cementing property, which improves the compression resistance and freezing resistance of the stabilized macadam of base course.

The calculated loss rate of compressive strength is shown in Figure 3. The strength loss after freeze-thaw cyclic tests decreased as the content of geopolymer continuously increased. The strength loss reduced from 14.2% to 2.5% when the dosage increased from 2% to 5%.

**Table 5.** Results of freeze-thaw test.

| Freeze-Thaw Cycle No. | $R_C$ before Freezing | $R_{DC}$ after Freezing | BDR (%) |
|---|---|---|---|
| CFA-4 | 12.61 | 11.94 | 94.8 |
| GFA-2 | 9.85 | 8.45 | 85.8 |
| GFA-3 | 12.75 | 11.42 | 89.6 |
| GFA-4 | 15.40 | 14.84 | 96.4 |
| GFA-5 | 16.72 | 16.30 | 97.6 |

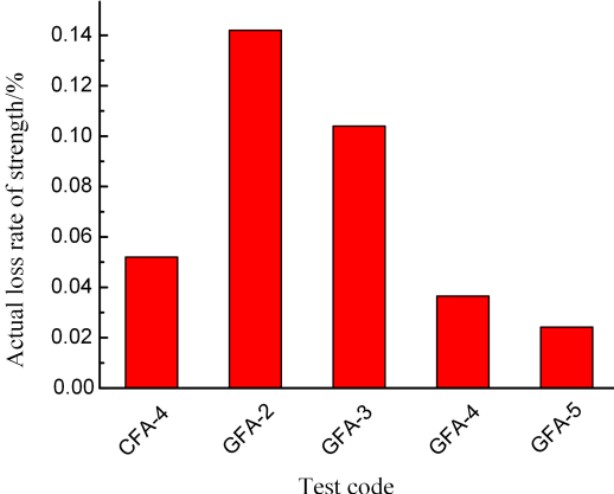

**Figure 3.** Strength loss of different binder stabilized macadam base materials after freeze-thaw cyclic tests.

*3.4. Dry Shrinkage Test*

The effect of geopolymer dosage on dry-shrinkage properties of stabilized macadam is shown in Table 6.

According to the data in Table 6, with increased dosage of geopolymer content, the maximum water loss, the maximum dry shrinkage strain, and the average dry shrinkage coefficient of the test pieces are all increased. When the mix proportion is same, the average dry shrinkage coefficient of slag/fly-ash-based geopolymer-stabilized macadam is slightly smaller than that of cement-fly-ash-stabilized macadam. The optimal water content of stabilized macadam is the highest with the largest dosage of geopolymer. Most water evaporation arises from the free water in test pieces at the early stage, which causes relatively greater water loss.

**Table 6.** Results of dry shrinkage.

| Test Code | Length of Test Piece (mm) | Maximum Water Loss (%) | Maximum Dry Shrinkage Strain (με) | Average Dry Shrinkage Coefficient (με/%) | Initial Mass (g) | Constant Weight after Drying (g) |
|---|---|---|---|---|---|---|
| GFA-2 | 400 | 4.01% | 187.1 | 46.7 | 9866.6 | 9486.1 |
| GFA-3 | 400 | 4.24% | 231.2 | 54.5 | 9830.9 | 9431.0 |
| CFA-3 | 400 | 4.29% | 261.3 | 60.9 | 9868.3 | 9462.4 |
| GFA-4 | 400 | 4.53% | 321.4 | 70.9 | 9500.7 | 9089.3 |

Figure 4a shows the relationship between the water loss rate and curing ages of the four groups of test pieces. The water loss rate increased with increasing of curing ages. It increases significantly within the first 15 d, and then becomes slight. The water loss rate of GFA-2 test piece reached 3.44% in the first 15 d; the water loss rate of the GFA-3 test piece reached 3.67% in the first 15 d; the water loss rate of the CFA-3 test piece reached 3.78% in the first 15 d; the water loss rate of the GFA-4 test piece reached 3.9% in the first 15 d. The water loss rates of the four groups of test pieces in the first 15 d accounted for about 85% of the total water loss rate, indicating that most water loss occurred at the early stage.

Figure 4b shows the curve of the relationship between dry shrinkage strain and curing ages of the four groups of test pieces. The dry shrinkage strain of test pieces is also increasing as curing ages increase, which is almost consistent with the change in water loss rate. It increases rapidly in the first 15 d, which further proves that the evaporation of water in test pieces lead to drying shrinkage of the interior of test pieces.

The change of cumulative dry shrinkage coefficient with curing ages in each group of test pieces is shown in Figure 4c. The accumulative dry shrinkage coefficient increases rapidly at the early stage due to the rapid water loss. Then, the increase rate slows down. The dry shrinkage coefficient first constantly increases but then decreases as the water loss of the test pieces constantly reduces.

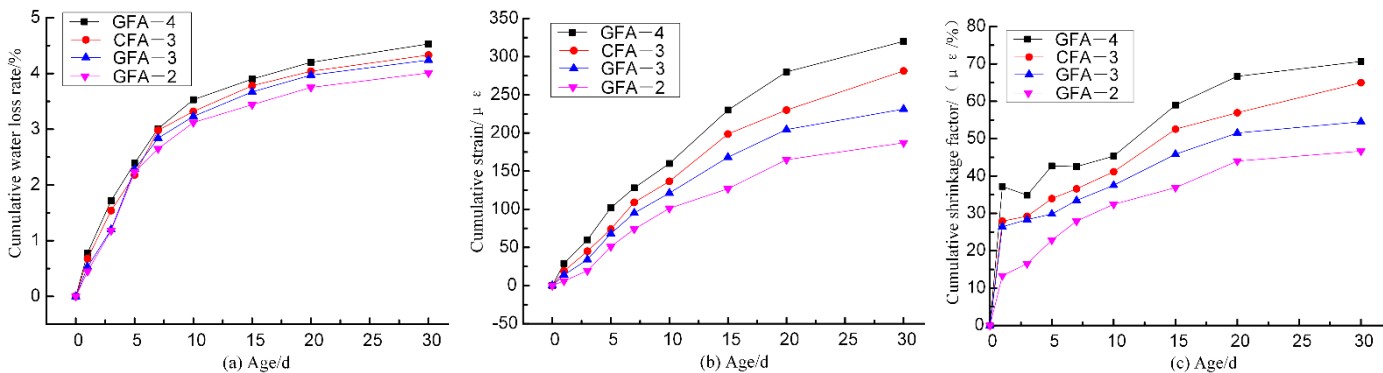

**Figure 4.** Dry shrinkage test change with time. (**a**) Cumulative water loss rate. (**b**) Cumulative strain. (**c**) Cumulative shrinkage factor.

### 3.5. Microscopic Test

3.5.1. SEM Analysis

Figure 5a,b show the SEM images of 28 d slag/fly-ash-based geopolymer-stabilized macadam (GFA-4). It can be observed that the slag/fly-ash-based geopolymer produces many gelatinous substances with a dense and continuous irregularly shape, which significantly contributes the strength of slag/fly-ash-based geopolymer. Larger unbroken or incompletely broken spherical fly ash particles can be observed, which can decrease the reaction rate. Furthermore, the geopolymer consisted of a C-S-H, C(N)-A-S-H gel-generated tetrahedral network structure due to the subsequent hydration and polymerization reaction in alkaline environment. The residual quartz may act as an inert filler to fill micro-voids, thus improving material stability [27–29]. More silicon dioxide participates in polymerization and produces more calcium silicate hydrate due to the addition of fly ash [30].

The interfacial transition zone (ITZ) of slag/fly-ash-based geopolymer stabilized macadam (GFA-4) after 28 d of curing is shown in Figure 6a,b. The result shows that no obvious porous around the ITZ was detected, while there is intrusion or overlay of reaction products with the surface of aggregates, indicating that the slag/fly-ash-based geopolymer and aggregates were well bonded. At the same time, a large amount of fly ash residues was found near the ITZ. This is probably because the adsorption of cementitious materials on the surface of aggregates causes loose and porous microstructure near the interface, which results in an uneven strength and affects the integrity of the structure [15].

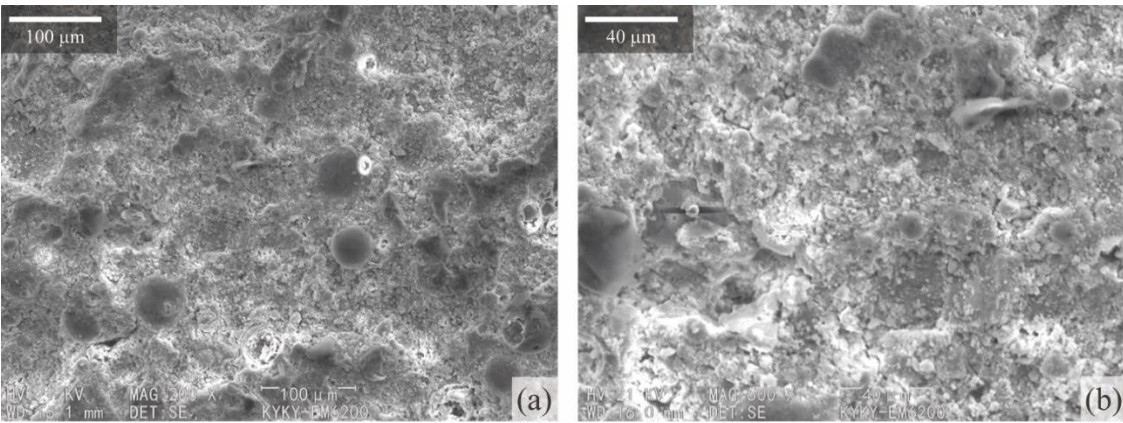

**Figure 5.** SEM images of slag/fly-ash-based geopolymer stabilized macadam (**a**) 100 μm. (**b**) 40 μm.

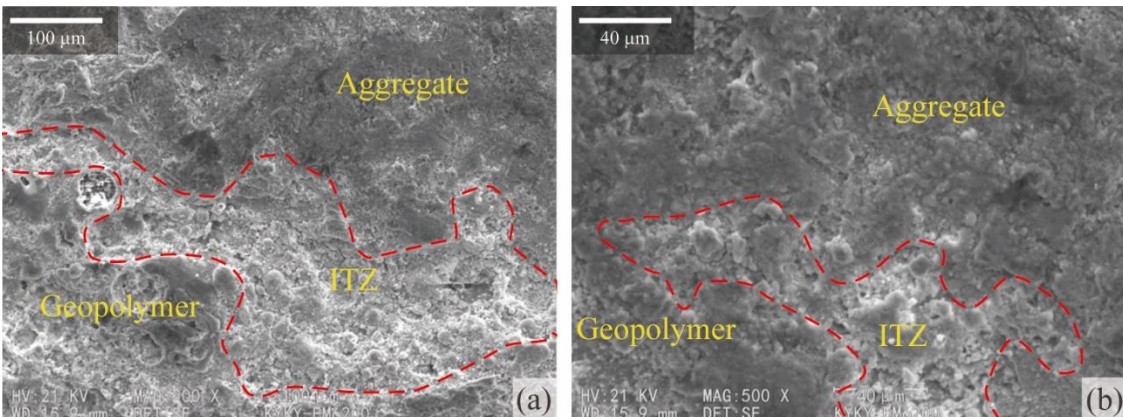

**Figure 6.** ITZ images of slag/fly-ash-based geopolymer-stabilized macadam. (**a**) 100 μm. (**b**) 40 μm.

### 3.5.2. FTIR Analysis

Figure 7 shows FTIR images of slag/fly-ash-based geopolymer-stabilized macadam (GFA-4) after curing for 7 d and 28 d. The two characteristic peaks appeared near 1500 cm$^{-1}$ and 3500 cm$^{-1}$ represent the absorption bands of water molecules. The characteristic peak near 1500 cm$^{-1}$ is vibration peak of H-O-H, and the characteristic peak near 3500 cm$^{-1}$ is the vibration peak of O-H. The former is mainly related to free water, while the latter is related to crystal water and free water in the test piece. Therefore, both adsorbed water and crystal water exist in the test piece [31].

The characteristic peak near the wavelength of 1000 cm$^{-1}$ is the symmetric stretching vibration absorption peak of Si-O-Al in the geopolymer. It is found that the strength of the 28 d test piece is stronger than that of the 7 d test piece because the hydration reaction is relatively adequate along with the formation of more aluminosilicates as the curing age prolonged. At the same time, glass phases are constantly dissolved in the alkaline solution to generate more silica-alumina tetrahedral network structures, thus improving the strength [32,33].

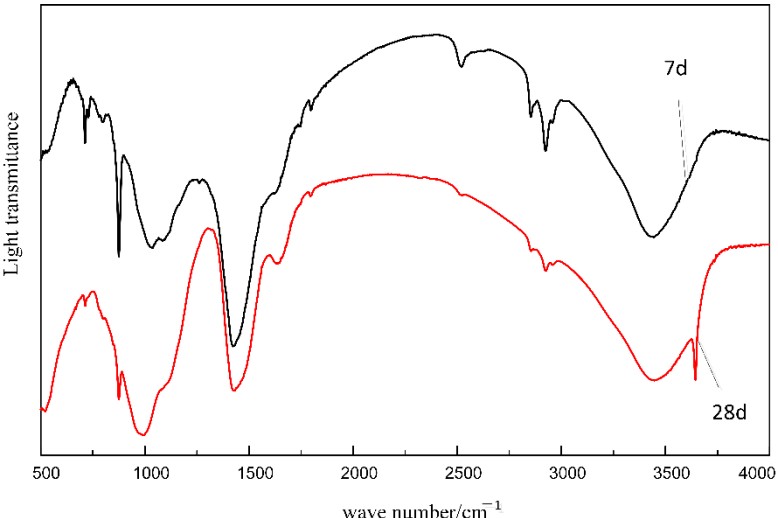

**Figure 7.** Infrared spectrum of slag/fly-ash-based geopolymer-stabilized macadam.

### 3.5.3. XRD Analysis

Figure 8 shows XRD of slag-based geopolymer and slag-based geopolymer added with fly ash (GFA-4) after curing for 28 d. The main components of slag are CaO and $SiO_2$. A variety of gel phases are produced under the action of alkali activators due to high content of Ca and low content of Si and Al, among which calcium-rich phase products, including calcium carbonate, account for the majority. Calcium carbonate is generated due to carbonation of slag after contacting with air. In addition, diffraction peaks are generated near 27° and 31°, indicating that it also contains zeolites ($NaAlSi_3O_8$, $CaAl_2Si_2O_8$) and C-A-S-H, N-A-S-H gels. In addition, a large plenty of unreacted $Ca(OH)_2$ and quartz crystals are also identified. After addition the fly ash containing a large amount of $Al_2O_3$ and $SiO_2$, it is found that the unreacted $Ca(OH)_2$ in the system was consumed considerably. As a result, the intensity of the diffraction peaks of the generated hydrated calcium silicate, hydrated aluminosilicate, and alkali calcium aluminate crystal improved. This indicates that the hydration and polymerization reaction are improved [34,35].

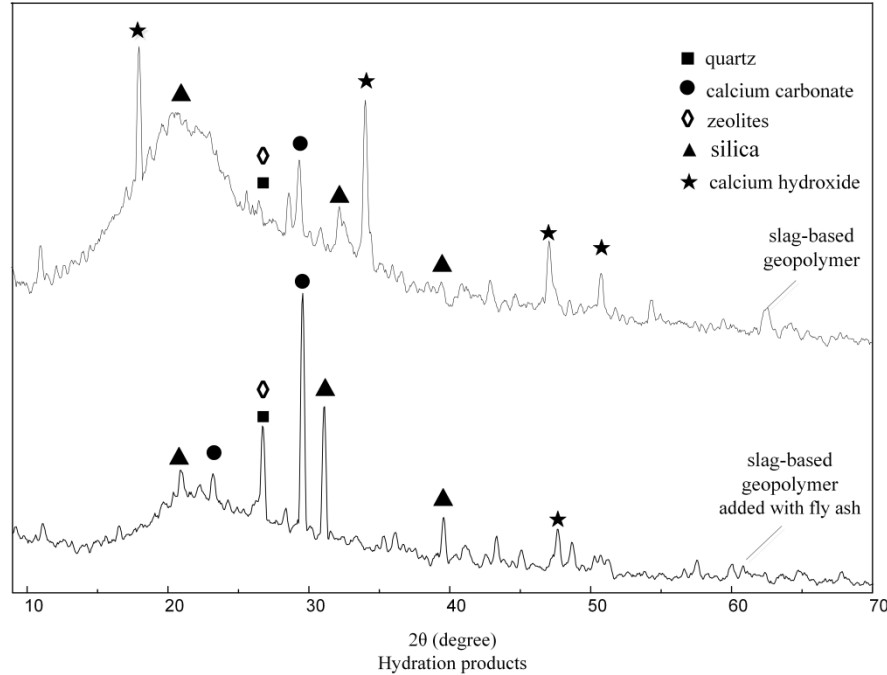

**Figure 8.** XRD patterns of slag-based geopolymer and slag-based geopolymer added with fly ash.

## 4. Conclusions

In this paper, the effects of curing ages and dosage of slag-based geopolymer on mechanical properties, freezing resistance, and dry shrinkage are carried out on geopolymer-stabilized macadam. A detailed analysis is performed for the results of mesoscopic test including XRD, FITR, and SEM. The main conclusions are as follows:

(1) Under the skeleton-density grading, when the ratio of slag-based geopolymer, fly ash and macadam is 4:16:80, the unconfined compressive strength of slag/fly-ash-based geopolymer-stabilized macadam reached 8.76 MPa after 7 d of curing. It further reached 15.4 MPa after 28 d of curing. Therefore, it can satisfy the compression strength requirements of expressway and first-class highway base (JTG/T F20-2015).

(2) Through data fitting, it is found that the compressive strength, elastic modulus, and tensile strength of 28 d geopolymer-stabilized macadam are all in a linear relationship with the dosage of geopolymer and increase linearly with increased dosage.

(3) The freeze-thaw resistance of slag/fly-ash-based geopolymer-stabilized macadam is better than that of cement/fly-ash-based stabilized macadam. With increased dosage of geopolymer, the strength loss after freezing and thawing decreases from 14.2% to 2.5% when the dosage increases from 2% to 5%. In addition, the dry shrinkage performance of slag/fly-ash-based geopolymer-stabilized macadam is also better than that of cement/fly-ash-based stabilized macadam. The dry shrinkage strain of slag/fly-ash-based geopolymer-stabilized macadam with 3% content is 231.2 $\mu\varepsilon$, which is slightly smaller than that of cement/fly-ash-based stabilized macadam (261.3 $\mu\varepsilon$).

(4) The reaction of geopolymer cementitious material itself generates stable products with higher strength. After addition of fly ash, a large amount of unreacted $Ca(OH)_2$ is consumed, the hydration and polymerization reaction is improved, and the bonding between slag/fly-ash-based geopolymer and aggregates is satisfactory.

**Author Contributions:** Conceptualization, Y.H. and J.Y.; methodology, J.Y.; software, X.N.; validation, X.N., Z.W. and J.L.; formal analysis, Z.W.; investigation, J.L.; resources, Z.W.; data curation, J.L.; writing—original draft preparation, X.N.; writing—review and editing, Y.H.; visualization, X.N.; supervision, Y.H.; project administration, Y.H.; funding acquisition, J.Y. All authors have read and agreed to the published version of the manuscript.

**Funding:** This research was funded by the Project of Science and Technology of Henan Transportation Department, grant number 2018G11 and grant number 2022-5-1. At the same time, this research was also funded by the Key Science and Technology Project of Henan Province, grant number 212102310937.

**Institutional Review Board Statement:** Not applicable.

**Informed Consent Statement:** Not applicable.

**Data Availability Statement:** Details on all data supporting the reported results can be obtained in Tables 1–6 and Figures 1–8 in this original manuscript.

**Conflicts of Interest:** The authors declare no conflict of interest.

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
