# Peer review of "Research on the Pavement Performance of Slag/Fly Ash-Based Geopolymer-Stabilized Macadam"

_applsci, doi:10.3390/app121910000_

Round 1

Reviewer 1 Report

I do not have any comments. It is interesting and very well organised paper. 

Author Response

Thank you for your review.

Reviewer 2 Report

The study examined the application of slag/fly ash-based geopolymers in macadam base. The experimental design was clear and findings were interesting. Some comments were given as follows:

1 In Table 3, the author set six type of geopolymer (CFA-3 & 4 and GFA 2 to 5). However, the result and discussion did not show full comparison and in-depth discussion. Why did not provide the heavy compaction test for CFA-3 & 4 in section 3.1? Please also explain why only lists CFA-4 in section 3.2 and 3.3; CFA-3 was only used in section 3.4.

2 Please define JPA listed in Table 3. What is about the specification?

3 Please list a table for compassion of the properties (e.g. section 3.1-3.4) of geopolymer and JTG/T F20-2015 and/or other previous studies to support the conclusion.

Author Response

Dear reader:

This is my reply about your question, please review.

Reviewer 3 Report

Please have a look on the attached file, I have two concerns 

Author Response

Dear reader:

read circles have been deleted, Aggregate and ITZ as well as Aggregate and Geopolymer have been re-marked with a dashed red line.

Round 2

Reviewer 3 Report

Thanks for addressing the comments positively all done